# Large language model based framework for automated extraction of genetic interactions from unstructured data

Jaskaran Kaur Gill[1]*, Madhu Chetty[1]*, Suryani Lim[1], Jennifer Hallinan[1,2]

1 Health Innovation and Transformation Centre, Federation University, Ballarat, Victoria, Australia,
2 BioThink, Brisbane, Queensland, Australia

* jaskarankaurgill@students.federation.edu.au (JKG); madhu.chetty@federation.edu.au (MC)

**Data Availability Statement:** The GIX framework code, pseudocode, datasets, extracted relations, models, and relevant data used in this study are available on GitHub at https://github.com/JaskaranKaurGill1/Gene-Interaction-Extraction.

## Abstract

Extracting biological interactions from published literature helps us understand complex biological systems, accelerate research, and support decision-making in drug or treatment development. Despite efforts to automate the extraction of biological relations using text mining tools and machine learning pipelines, manual curation continues to serve as the gold standard. However, the rapidly increasing volume of literature pertaining to biological relations poses challenges in its manual curation and refinement. These challenges are further compounded because only a small fraction of the published literature is relevant to biological relation extraction, and the embedded sentences of relevant sections have complex structures, which can lead to incorrect inference of relationships. To overcome these challenges, we propose GIX, an automated and robust **G**ene **I**nteraction **Ex**traction framework, based on pre-trained Large Language models fine-tuned through extensive evaluations on various gene/protein interaction corpora including LLL and RegulonDB. GIX identifies relevant publications with minimal keywords, optimises sentence selection to reduce computational overhead, simplifies sentence structure while preserving meaning, and provides a confidence factor indicating the reliability of extracted relations. GIX's Stage-2 relation extraction method performed well on benchmark protein/gene interaction datasets, assessed using 10-fold cross-validation, surpassing state-of-the-art approaches. We demonstrated that the proposed method, although fully automated, performs as well as manual relation extraction, with enhanced robustness. We also observed GIX's capability to augment existing datasets with new sentences, incorporating newly discovered biological terms and processes. Further, we demonstrated GIX's real-world applicability in inferring *E. coli* gene circuits.

## Introduction and motivation

The scientific literature holds significant biological insights about issues such as disease-causing mutations, health intervention methods, genome analysis, and potential drug targets. In this research, we focused on the inference of transcriptional regulatory behaviour by identifying genome-wide relationships among macromolecular entities such as genes, proteins, and RNAs [1]. The identification of such relations is relevant in developing accurate computational

**Funding:** The author(s) received no specific funding for this work.

**Competing interests:** The authors have declared that no competing interests exist.

models of genetic networks, providing insights into disease progression, intracellular functions, and drug behaviour.

The automated acquisition of prior knowledge from unstructured text is crucial but challenging, given the exponential growth in the published literature due to recent technological advancements in scientific research [2]. Medline (PubMed) reported a doubling of indexed articles on Gene Regulatory Networks in the last decade, reflecting increased interest in understanding genomic interactions at a cellular level, both experimentally and computationally [3]. Despite the huge number of papers available, only those with experimentally observed transcriptional links are relevant to the extraction of relationships among biological entities. Further, only sections containing key sentences can be used for identifying biological interactions. These sentences reference multiple biological entities, and have complex structures comprising several clauses and technical terminology [4]. Biological experiments are susceptible to errors during data collection, demanding substantial efforts to establish the relevance and authenticity of extracted regulations. Thus, most structured databases such as TRANSFAC [5] and RegulonDB [6], rely on human experts to pre-process scientific papers, extract domain-related relations, and authenticate the information [7].

Manually curating genomic repositories and annotating genetic relationships from literature is time-consuming and challenging [8]. Traditional reading and summarising lack scalability, are subject to personal interpretations, and are time-consuming. Automating the pre-processing of scientific literature and post-processing of extracted information using Natural Language Processing (NLP) techniques can address these limitations. Several researchers have studied automating pre-processing and post-processing for accurately extracting relations from unstructured data. Most pre-processing techniques involve the extraction of all available abstracts within a specific timeline from online resources such as PubMed [9–11]. However, these steps, while effective in domains with simple concepts and interactions, prove inefficient in the complex biological domain, with entity relations specific to organisms, cell function, or disease conditions.

In the relation extraction (RE) stage following pre-processing, statistical methods aim to capture entity relationships through feature extraction [1]. Approaches such as sentence simplification [12], link grammar parsing [13], and a combination of vector and tree-based kernels [14] enhance protein-protein interaction (PPI) relation extraction in simpler sentences. However, these methods face limitations in handling complex biological sentences, and simplification can lead to overfitting and information loss. These methods heavily depend on tools like MedPost (a POS tagger) or existing biological knowledge bases. Fundel *et al.* [15] reported that most of the false interactions predicted by RelEx, a rule-based parse tree NLP technique, are due to incorrectly tagged entities, POS-label error, or insufficient rule-based sentence construction. Recently introduced pre-trained neural network models provide substantial benefits when capturing both syntactic and semantic information, surpassing the capabilities of machine learning methods and word embedding techniques [16–18]. Despite the specific biomedical training of BioBERT (a pre-trained Bert-based model), its improvement in relation extraction tasks is marginal [19]. The complex sentence structure describing causal dependency still limits the applicability of RE techniques. Despite rigorous pre-processing and RE, inferred interactions may still contain noisy entities, inaccurate predictions, and conditional relations. A post-processing step such as unsupervised clustering [20] and confidence level calculation [21], if integrated with prior knowledge and model prediction confidence, can further refine results by eliminating falsely predicted named entities and relations.

Researchers have tackled different stages of relation extraction from the literature, including pre-processing, post-processing, and relation extraction itself. However, there is no overarching framework that comprehensively addresses all aspects of biological relation extraction.

While specific tools and technologies may evolve or change over time, a well-designed framework offers a stable underlying structure, methodology, or approach. To this end, we propose systematic, automated and robust Gene Interaction Extraction (GIX), that efficiently identifies relevant publications with minimal keywords, optimises sentence selection, and provides a confidence factor for the reliability of extracted biological relations. GIX is a novel three-stage framework for Gene Interaction Extraction, involving: (i) Pre-processing; (ii) Relation extraction; and (iii) Post-processing. Leveraging the fine-tuned pre-trained domain-specific NLP models, GIX is designed to optimally extract regulatory links among genes/proteins in a set of abstracts from PubMed. It automates all of the processes required for biological RE, and eliminates the manual intervention required to gather and transform problem-specific data into an acceptable RE format. The pre-processing phase involves an abstract search for target-network information, selecting sentences on transcriptional regulation to eliminate irrelevant text, reduce computational overload, and improve accuracy. In the Relation Extraction stage, we harness the biological contextual understanding of BioBERT and BERN2 (a pre-trained Bert-based Bio-named entity recognition model). Additionally, an entity-labelling schema is proposed to enhance the accuracy of relation prediction. This schema works by reducing sentence complexity without compromising grammatical structure, avoiding information loss. In post-processing, BERN2 and tailored NLP techniques refine extracted biological relational entity names, incorporating a novel confidence measure to authenticate regulations and sources and improve accuracy by eliminating false positives.

Several experiments were performed using datasets including BioInfer [22], HPRD50 [15], IEPA [23], and LLL [24]. The proposed RE model (Stage-2 of the framework) achieved a significant improvement of 13.7% in F-Score for HPRD50 compared to the previous best-performing method. GIX, achieved superior performance in RE from dataset LLL and database RegulonDB. In experiments with LLL, our results showed that our GIX framework not only achieved optimum accuracy but also reported multiple relation dictating sentences per regulation, as opposed to the single-sentence per relation manner common in benchmark datasets.

The structure of the paper is as follows: an overview of the relevant preliminaries is provided in the "Background" section. The "Methods" section details the processes and models used in our three-staged GIX framework. In the "Results" section, we cover the experimental setup of benchmark datasets, present their respective results, and discuss the outcomes. Lastly, the "Conclusion" section concludes the paper and explores future directions.

## Background

### Unstructured text and relation extraction

Structured data, such as tables and databases, have a consistent layout and predictable pattern. In contrast, unstructured data such as written and spoken text lacks a predefined structure, making it difficult to process [25]. These data can consist of diverse languages, and contain grammatical errors, abbreviations, and context-dependent meanings. It is difficult to retrieve only relevant data, because of the size of datasets, the diversity of publications, and the rapid evolution of multidisciplinary fields. Efforts to standardise vocabulary in biomedical literature, such as MeSH, help tackle unstructured data by assigning terms, aiding in information retrieval and relation extraction, thereby enhancing biomedical research. RE is more challenging than named entity recognition or classification, due to the lack of explicit markers for relationships and the complex contextual dependencies between entities, which make it hard to accurately identify and extract the underlying relationships from unstructured text. RE tasks can be classified into one of two categories: (1) rule-based methods which identify pre-defined patterns; and (2) machine-learning (ML) models which treat RE as a classification problem

[26]. ML based RE approaches have been further classified into kernel-based, feature-based, and deep learning (DL) categories. Various ML methods have been used to extract regulations among genes and gene products However, building and training RE models is a complex and time-consuming task, as the models need to be trained using a large dataset. Advanced NLP techniques, including text mining, RE, and named entity recognition, have proved very effective in extracting meaningful information from complex and variable unstructured text [27].

## Pre-trained large language models

In recent years, pre-trained models based on NLP techniques have been shown to work effectively with unstructured text, and have been used in a range of applications. Being pre-trained on large data, they require little or no training. Well known large language models (LLM) such as Embeddings from Language Models (ELMO) [28] and Bidirectional Encoder Representations from Transformers (BERT) [29] are pre-trained on massive amounts of text data, allowing them to learn rich linguistic patterns and semantics. ELMO introduced contextual word embeddings, in which each word representation is dynamically generated based on the surrounding context. BERT is a transformer-based model that learns bidirectional contextual representations of words. Both ELMO and BERT models have demonstrated excellent performance across various NLP tasks, including named entity recognition, part-of-speech tagging, sentiment analysis, and RE [30–33]. However, BERT has been shown to surpass ELMO in performance, as it learns bidirectional contextual representations, enabling a deeper understanding of semantic relationships [34]. The domain-specific LLMs perform RE by capturing specific knowledge, thus improving accuracy. BioBERT, the BERT model optimized for biomedical text mining, is domain-specific, and is pre-trained on PubMed abstracts and PMC articles [19]. BioBERT can easily be fine-tuned to perform text extraction tasks, and has been successfully applied to tasks such as the allocation of phenotypes to protein-protein interactions and the extraction of drug-drug interactions [35,36]. It has outperformed other state-of-the-art NLP-based RE methods in the biological domain [19].

## Named entity recognition

Named entity recognition (NER) and biological NER (BioNER), identifies biological entities such as genes, proteins, diseases, drugs, and miRNAs [37]. BioNER methods fall into one of three categories: (1) knowledge-based; (2) rule-based; or (3) machine learning [38]. A knowledge-based approach uses an existing database or dictionary to identify known entities. Such methods are simple to implement, but limit NER tagging to known entities [39]. The rule-based approach tends to overfit and fails to generalise, and thus is ineffective when applied to all cases. State-of-the-art machine learning techniques use POS tags and apply grammatical structure and interdependencies within a text to conveniently identify the named entities. Gene and protein names have been labelled using a combination of conditional random fields (CRF) and bidirectional long short-term memory (LSTM) architecture [40]. ML models such as Support Vector Machines (SVMs) and hidden Markov models have been used for BioNER. Although ML techniques perform better than other traditional methods, they require a large amount of manually annotated training data [38]. Pre-trained language models such as Bio-BERT can be fine-tuned to easily perform BioNER without needing a large amount of training data. BERN2, a BioNER tool reported in 2022, not only supports NER, but also allows named entity normalization (NEN). NEN allows mapping recognised named entities to a common or canonical form ensuring uniform representation of named entities. BERN2 has outperformed existing BioNER tools, including BERN, on several applications, including the identification of diseases, drugs, species, genes, and proteins.

## Datasets

Some of the comprehensive repositories of curated protein and genetic interactions collected from the scientific literature are BioInfer [22], HPRD50 [15], IEPA [23], and LLL [24]. These resources are commonly used as benchmark datasets with which to study RE [15,41,42]. BioInfer contains a total of 9,666 full dependency annotations of gene, protein, and RNA regulations from 1,100 sentences. HPRD50 includes 50 abstracts from Human Protein Reference Database (HPRD) for direct regulation relation annotation. IEPA, is an Interaction Extraction Performance Assessment of 300 abstracts using two named biochemical entities. LLL, the Learning Language in Logic challenge, contains 330 gene/protein interactions labelled as regulatory from 77 unique sentences. Processing these datasets for RE requires a combination of advanced NLP techniques, domain knowledge, and careful pre-processing to handle the wide range of interaction types, varying sentence structures, and the need to disambiguate entities and their relationships.

RegulonDB is a manually annotated, publicly available database containing transcriptional regulations in *Escherichia coli*, also known as *E. coli* [6]. The database contains transcription factor (TF) regulations including TF-gene, TF- transcriptional unit, TF-operon, and TF-TF. Each regulation is classified as weak or strong based on the type of experiment used to identify the interaction. For instance, a ChIP analysis with statistical validation is considered to provide stronger evidence than ChIP-chip only or ChIP-sequence analysis.

## Methods

This section addresses the improvement of the prediction accuracy of genetic interactions using the published literature. We propose an automated framework involving two Large Language Models (LLM)—BioBERT and BERN2—which are pre-trained on a large corpus of biological data. The overall architecture of the proposed LLM based framework is presented in Fig 1. The framework has three stages: (i) Stage-1: Pre-processing; (ii) Stage-2: Relation extraction; and (iii) Stage-3: Post-processing. The pre-processing and post-processing stages in GIX play a crucial role in addressing the inherent imbalance in biological datasets. In a sentence with 2 entities, the number of potential relationships, taking direction into consideration is

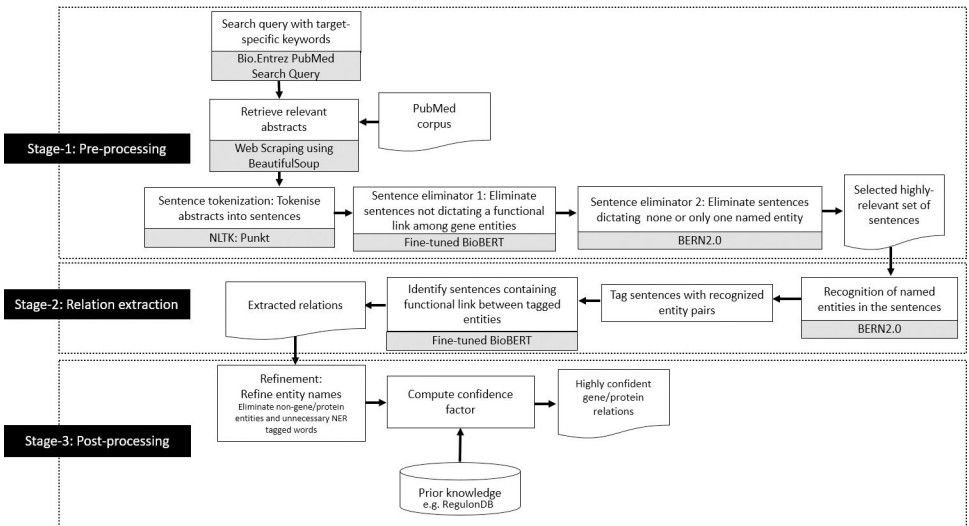

**Fig 1. Schematic of fully automated relation extraction using an LLM based GIX framework.** The white blocks signify the processes, and the grey blocks indicate the tools utilised within those processes.

two. As the number of entities in a sentence increase, the possible relationships amongst them can grow exponentially. However, the number of true relationships does not grow at a similar pace, thereby, leading to a situation when the number of non-interactions far exceed the number of interactions and resulting to imbalance dataset. The pre-processing (Stage-1) and post-processing stages (Stage-2) in GIX are designed to recognise non-relation entity pairs, to decrease the number of sentences classified as true negatives. This reduces the gap between the two classes, which contributes to the imbalance, thereby enhancing the overall performance and accuracy of relation extraction. We begin with the explanation for each component of Stage-1.

## Pre-processing

In Stage-1, we pre-process the input publications to extract only highly relevant text. This task is challenging because genomic transcriptional regulations can differ between organisms, and molecular entities may behave differently under different cell conditions. We formulated a search criterion which ensured the extracted literature's association to the specified conditional parameters. Known attributes of the target network, such as the organism and cell function, were included to in narrow down the search to closely related literature. As published papers often include a keywords section that helps indexers, these were also used to establish relevance. Multi-cellular organisms may exhibit different regulatory relations under different cell conditions; thus, the use of additional criteria can significantly improve the quality of obtained literature, thereby improving the accuracy and relevancy of extracted relations. The chosen set of keywords used for this study is outlined in the experiment section under "Selection of Keywords."

The Bio.Entrez module facilitates data retrieval from PubMed, a comprehensive biomedical literature database [43]. To find relevant PubMed articles, we crafted a search query using PubMed Bio.Entrez utilities. The PubMed search query requires a set of keywords and specifies the maximum number of documents to retrieve, resulting in a list of PubMed IDs ranked by relevance according to the provided keywords. With the retrieved PubMed IDs, we used the web scraping tool BeautifulSoup [44], a Python library, to extract titles and abstracts from the papers on PubMed. The set of abstracts obtained using the selected keywords is further processed to eliminate those papers which do not contain a reference to the organism, either in its full form, or as abbreviations.

**Sentence tokenization.** We split the final set of abstracts into individual sentences using Punkt sentence tokenizer from the Natural Language Toolkit (NLTK) [45]. Punkt is a pre-trained unsupervised machine learning model designed for detecting sentence boundaries. Then the tokenized sentences undergo two consecutive sentence elimination processes: the first utilises a fine-tuned BioBERT (biobert_v1.1_pubmed) classification model, and the second involves BERN2, as detailed in the subsequent sections.

**Sentence eliminator 1.** Sentence eliminator 1 identifies and removes sentences that do not discuss a regulatory interaction. A fine-tuning dataset was created from annotated PPI corpus sentences. The specific datasets used for fine-tuning and testing in each experiment are outlined in the results sections. To create the fine-tuning dataset, sentences dictating at least one regulatory relationship receive a positive classification label, while the rest are labelled as negative. Once finetuned, the BioBERT model evaluates test sentences, assigning a classification of 0 if there are no genetic interactions or if the sentence discusses non-regulation, and 1 if the sentence contains a relational context of a gene/protein interaction. Sentences classified as 0 are eliminated, while those classified as 1 undergo further evaluation for the presence of named entities.

**Sentence eliminator 2.** For the second sentence elimination step, since at least two entities are involved in a regulatory relationship, all sentences that contain fewer than two gene or protein entities are excluded. This elimination is carried out using BERN2, without any additional fine-tuning, to recognise gene and protein entities in a sentence. BERN2 can be implemented through their RESTFUL API [46]. For each sentence, BERN2 produces a JSON list of annotations, containing the entity phrase, its entity type, and a probability score. Sentences are eliminated if the annotation list contains fewer than two gene/protein entities.

## Relation extraction

In Stage-2 of GIX, we used BERN2 for NER and fine-tuned BioBERT for relation classification. Through fine-tuning on expert-annotated data, pre-trained models efficiently transfer general language knowledge to domain-specific tasks, resulting in improved task performance and adaptability. In this work, our goal was to extract binary relations according to their suitability for comparison, interpretability, and scalability. To extract binary relations, each entity pair in a sentence is substituted with a label to clearly identify both the agent and target entities involved in the relationship. Sentence eliminator-2, filters out sentences with fewer than two gene/protein entities, so the sentences undergoing processing for relation classification invariably contain a minimum of two gene/protein entities, irrespective of the presence or absence of an actual relationship. The labelling criteria remain consistent for all sentences before relation extraction. To extract binary relations, each entity pair in a sentence is substituted with a label to clearly identify both the agent and target entities involved in the relationship. The first entity of a pair is replaced with $GENE_AGENT# and the latter is substituted with $GENE_-TARGET#. The selected entity labels are descriptive, unique, and ensure consistency. Despite the advantages of selected labels, the structural complexity and presence of multiple entities in biological sentences can hinder the sequence classifier model's ability to recognize the labeled entities. To address this issue, any entity in a sentence other than the current pair is replaced with the word "BLANK", so that the model can easily identify the pair in consideration during classification. For example, in Fig 2, three genes—*SigK*, *GerE*, and *ykvP*—appear in the sentence, and for the gene pair *Sigk* ($GENE_AGENT#) and *ykvP* ($GENE_TARGET#) the remaining third gene is labelled BLANK. BioBERT, pre-trained on biomedical data, has a contextual understanding of complex biological terms and is able to manage this style of labelling. While the selected labelling tags ($GENE_TARGET#, $GENE_AGENT#) in complex sentences may not effectively highlight the target of the classification task and thereby limit model performance, anonymizing the entities additional to the tagged pair using "BLANK" effectively suppresses the unwanted entity without altering the lexical and semantical structure of the sentence.

The output of the RE process is a set of entity pairs from BioBERT, with classification prediction values varying between 0 and 1.

## Post-processing

The extracted relations may still contain wrongly predicted interactions or entities, due to the biological complexity of the relationships. We cannot assume that all stated relations in

---

**Original sentence:** Both SigK and GerE were essential for ykvP expression

**Tagged sentence:** Both $GENE_AGENT# and BLANK were essential for $GENE_TARGET# expression

---

**Fig 2. Illustration of NER tagging in sentences with multiple gene or gene products.**

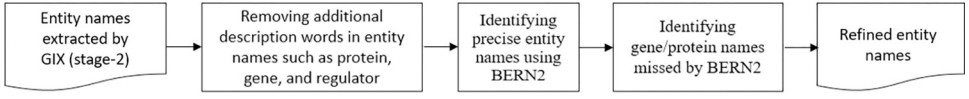

**Fig 3. Entity name refinement.**

published text are necessarily correct. Positive results are more likely to be published than negative or inconclusive results. This bias can lead to an overrepresentation of certain findings, which can skew the domain knowledge. Studies with small sample sizes, which are not representative of a larger population, may contain results that are unreliable or not generalizable. Additionally, even though our entity labelling schema aims to identify the controlling ($GENE_AGENT#) and child ($GENE_TARGET#) entities in a relationship, not all sentences clearly indicate the direction of the interaction. In such cases, information about known controlling entities, such as regulatory genes or transcriptional factors, in the target network can help identify the direction of the relation. In the post-processing stage, we therefore evaluate the *trueness* of each extracted relation based on several factors, including whether the relation was extracted from multiple documents, its existence in online repositories, and whether it involves a known regulator.

**Refinement.** The entire three-step refinement process is illustrated in Fig 3.

Before establishing well-known true regulations, the extracted relational entity names are refined. During NER, BERN2 may recognize a group of words or phrases as the named entity. For instance, consider the following sentence:

*"The Dnak suppressor protein interacts with molecular chaperones to assist in protein folding and prevent misfolding or aggregation."*

The phrase "Dnak suppressor protein" is identified as a gene/protein entity by BERN2, helping to reduce the structural complexity of the sentence. However, we need to extract only the entity name "Dnak" to successfully group regulations extracted from different research papers. To extract just the entity name, we split the phrases and process the individual words for NER using BERN2 (Fig 4). As depicted in example (ii) in Fig 4, the BERN2 tagged entity may not contain the entity name at all. Such entity relations are incomplete, and thus should be eliminated. For larger datasets, the use of BERN2 to process each word in an entity name can become computationally expensive. To address this issue, we created a list of the most repetitive non-entity words (available on the Gene-Interaction-Extraction GitHub repository

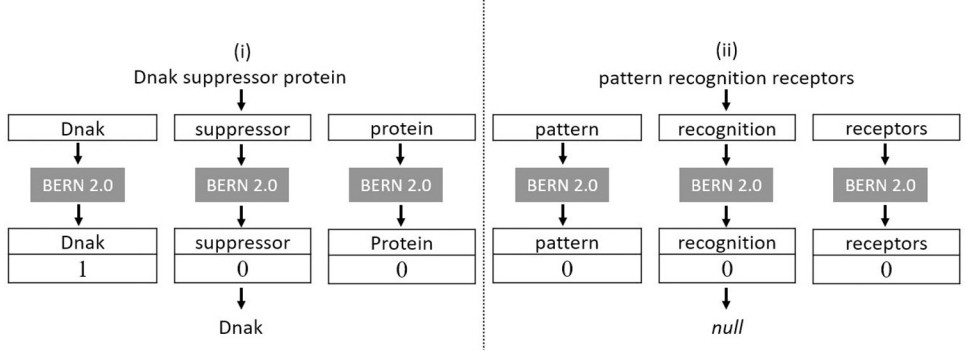

**Fig 4. Gene name refinement using BERN2.**

https://github.com/JaskaranKaurGill1/Gene-Interaction-Extraction) reflecting which non-entity words are most frequently removed before passing on to BERN2.

If BERN2 is unable to recognise the gene/protein name from a phrase without the adjoining words, the refined entities which return null names are checked to detect any missing genes or proteins. Most gene/protein names consist of letters, numbers, and symbols that may not conform to typical English language words, leading the spellchecker to flag them as potentially misspelled. Thus, the checking is done by using the word spell check tool from Textblob [47] to label individual words in entity names and identify incorrectly spelled words. If a word in an entity is incorrectly spelt, it is considered to be a protein or gene name, whereas if it is correctly spelt, it is assumed to be descriptive. The output of the refinement process is a set of entity-pairs composed only of gene or protein names.

**Confidence factor.**   We developed a new measure, the Confidence Factor $CF_{e_a e_t}$, to indicate the likelihood of existence of each of the interactions identified as existing between the agent/controller entity $e_a$ and the target entity $e_t$. For each entity pair $e_a e_t$, its corresponding $CF_{e_a e_t}$ is computed as follows.

$$CF_{e_a e_t} = \sum_{s=1}^{n} v_s + K \sum_{T=1}^{M} P_{e_a e_t}^{T} \tag{1}$$

$$= CF_1 + K * CF_2$$

$$P_{e_a e_t} = \begin{cases} 1, e_a \text{ is known agent but relation } e_a e_t \text{ is unknown} \\ 3, e_a e_t \text{ is known relation} \\ 0, \text{ otherwise} \end{cases} \tag{2}$$

Here, the variable $n$ denotes the total number of unique sentences obtained from GIX predicting $e_a e_t$, i.e. the regulation between $e_a$ and $e_t$. The variable $v_s$ is the RE classifier prediction of regulation $e_a e_t$, of the $s^{th}$ sentence. It is obtained as the Stage-2 output of the GIX and has a value between 0 and 1. The parameter $P_{e_a e_t}$ represents the prior knowledge about both the agent entity ($e_a$) and its interaction ($e_a e_t$) with the target entity $e_t$. The constant $M$ is the total number of curated databases under consideration. $K$ is a factor balancing the influence of terms $\sum_{s=1}^{n} v_s$ and $\sum_{T=1}^{M} P_{e_a e_t}^{T}$ (also referred to as $CF_1$ and $CF_2$). As shown in Fig 5, the discrete variable $P_{e_a e_t}$ of Eq 2 can acquire three different values for three different conditions, namely, (i) If $e_a$ is known to be a controller gene and the relation $e_a e_t$ is unknown, $P_{e_a e_t} = 1$; (ii) If $e_a e_t$ is a known relation, the value of $P_{e_a e_t}$ is given a higher value of 3 compared to (i) accounting for the presence of two entities and a connecting arc; (iii) For other conditions, $P_{e_a e_t} = 0$.

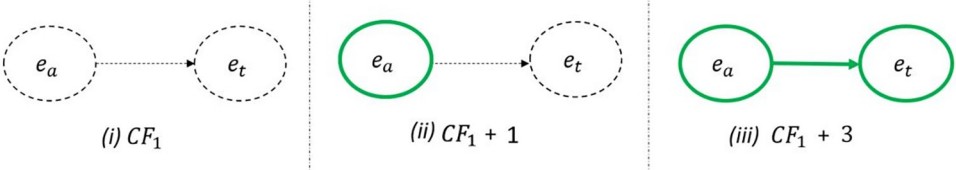

**Fig 5. Role of prior knowledge component CF$_2$ with respect to CF$_1$ on overall confidence factor $CF_{e_a e_t}$.** (i) No prior knowledge is available (ii) $e_a$ is a known controlling entity but $e_a e_t$ is an unknown relation (iii) $e_a e_t$ is a known relation.

A specific regulation can appear multiple times in each sentence. For example, in the sentence

*"The study found that the upregulation of TrpE was associated with an increased expression of trpR in the cell line, and this TrpE -dependent trpR expression initiates the enzyme activity."*

The relation between *TrpE* and *trpR* appears twice with two different values of $v_s$. In such situations, the average value of $v_s$ is considered. Previously curated annotations of a genetic interaction can also be used to validate the correctness of the GIX's extracted relation. The manually curated databases cite multiple published papers reporting genomic interactions. For instance, RegulonDB confirms the transcriptional interaction between TF CRP-cyclic-AMP and its target genes deoC, deoA, deoB, and deoD, from 16 sources. Similarly, interactions confirmed from a higher number of experimental and/or analytical sources will have a higher $CF_{e_a e_t}$ and thus, are more likely to be true. A threshold value ($CF_{e_a e_t} = \gamma$) is defined, and all regulations with $CF_{e_a e_t} < \gamma$ are treated as false positive. The final output of the framework will be a set of highly confident extracted entity pairs and their corresponding $CF_{e_a e_t}$ values, representing the accuracy of the retrieved relationships.

## Results

We first describe the experimental setup, including the hyperparameter configuration and evaluation metrics used. Subsequently, we discuss the selection of keywords for the extraction of relevant information from the target-related literature. We conducted three independent experiments to assess the effectiveness of GIX. The first experiment (Exp1) demonstrated the Relation Extraction Capability (Stage-2) of the GIX framework using four well-known benchmark datasets for gene/protein interactions. The second experiment (Exp2) evaluated automated extraction using GIX against the manual curation of a benchmark dataset. The third experiment (Exp3) evaluated GIX against the manual curation of a real-world database of transcriptional regulations. Finally, as a demonstration of the significance of GIX-extracted relations with their confidence factors, we used the relations for constructing gene regulatory networks.

### Experimental setup

We implemented our models using PyTorch transformers: an open-source library for machine and deep learning models [48]. The framework was written in Python 3.10.11 in Google Colab Pro. The hyper-parameters setup for the BioBERT model is given in Table 1. We trained our model on Google Colab using a GPU (Tesla P100-PCIE-16GB) with a BertAdam optimizer.

**Table 1. Hyperparameters used for the RE classification model.**

| Hyper-parameters | Value |
|---|---|
| Model | biobert_v1.1_pubmed |
| Token max length | 256/ 512 depending on the average length of sentences of the dataset |
| Optimizer | BertAdam |
| Batch Size | 8 |
| Number of epochs | 10 / 20 depending on the dataset size. |
| Learning rate (BertAdam) | 2e-5 |
| Warmup (BertAdam) | 0.1 |
| Learning rate decay (Weight decay rate) | 0.01 |

To assess the model performance, the metrics Recall (R), Precision (P), and F-score (F) were evaluated. Recall and Precision provide complementary insights into the performance of a model. Recall measures the proportion of relevant instances correctly identified, while Precision represents the accuracy of the model's positive predictions. The F-score provides a balanced assessment of a model's performance by considering both Precision and Recall, offering a single metric with which to evaluate classification accuracy.

## Selection of keywords

The choice of the correct set of keywords is crucial to GIX performance in target-related literature search. Our choice was based on predefining certain attributes of the required output. These attributes specify the type of relation being extracted, determining whether they pertain to a particular organism or a specific cell function. Incorporating these attributes makes the keyword selection process effective in focusing our search on finding literature directly related to specific aspects of the target network. We identified frequently used keywords in published papers related to genetic entity regulation and interaction. Fig 6 depicts the 20 most repeated keywords among papers used by RegulonDB for the manual extraction of the transcriptional relations of *Escherichia coli*. The name of the organism embodying the regulatory system is the most repeated keyword, and is thus included as one of the preferred keywords. Other important words—"*gene regulation*", "*gene expression*", "*transcriptome*", "*transcription factor*", "*regulation*", and "*posttranscriptional regulation*"—which are repeated frequently are also included as search terms. Thus, the selected set of keywords for this research is the combination of the common words "*gene regulation gene expression transcriptional*" and the name of the target organism.

## Exp1- Relation extraction capability (Stage-2) of the GIX framework

The performance of our RE with its improved entity-labelling schema (Stage-2 of the GIX framework, referred as GIX RE) was investigated using four well-known benchmark gene/protein interaction datasets: BioInfer, HPRD50, IEPA, and LLL. The distribution of positively and negatively annotated sentences for these four datasets is given in Table 2.

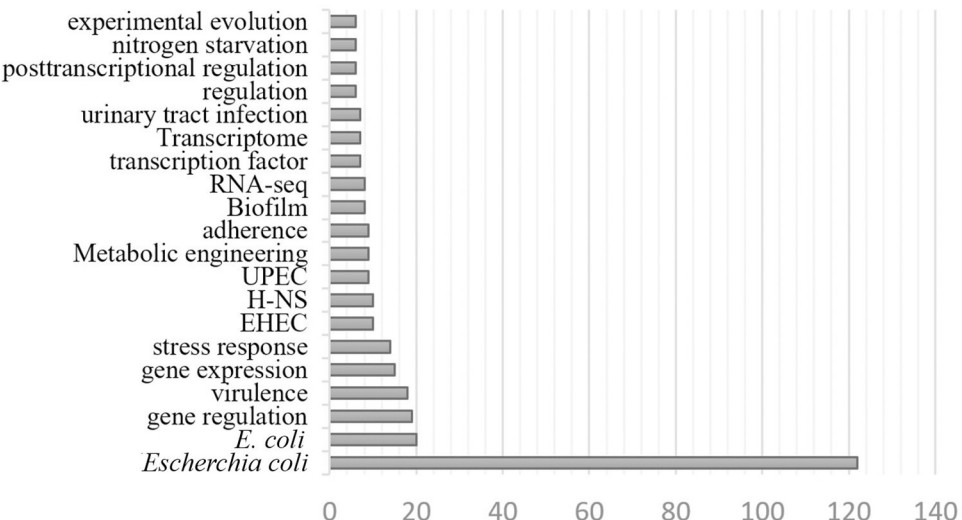

**Fig 6. Top 20 most repeated keywords included in papers referenced by RegulonDB for TF-binding sites.**

**Table 2. Distribution of positive and negative classifications in five benchmark PPI corpora.**

| Dataset | Positive | Negative | Unique Sentences |
|---|---|---|---|
| **BioInfer** | 2534 | 7132 | 1100 |
| **IEPA** | 335 | 482 | 486 |
| **HPRD50** | 163 | 270 | 145 |
| **LLL** | 164 | 166 | 77 |

To evaluate a model's performance and generalisation ability, we employed the widely used 10-fold cross validation provided by the KFold library from Scikit-Learn [49] to each of the four datasets (HPRD50, BioInfer, IEPA and LLL). In brief, the method involves dividing the dataset into 10 equal subsets, with 9 subsets used for fine-tuning and the isolated 10th subset used for testing. For each dataset, the process is repeated 10 times using a different fold as the test set, and each time, the relation classification model gets fine-tuned from scratch (original setting). The overall accuracy is determined by averaging the results from the 10 individual experiments (folds) conducted on each dataset (shown in Fig 7). This ensures comprehensive evaluation across diverse data samples, contributing to the model's robustness and generalisability. This approach has been commonly used in several state-of-the-art methods in different domains including Biological relation extraction,e.g. Bi-LSTM [50], MCCNN [51], GK [52], NHGK [53], EDG ([54], PIPE [42], WWSK [55], RCNN [56], DNN [57], RNN + CNN [56] and iLSTM+tAttn [41]. We used a token length of 256 for BioInfer, HPRD50, IEPA, and LLL. The smaller datasets, HPRD50, IEPA, and LLL, required 20 epochs for fine-tuning, whereas the larger dataset, BioInfer, achieved stability in just 10 epochs. The performance of the proposed RE with the improved entity-labelling schema (Stage 2 of GIX), compared with other state-of-the-art methods, is given in Table 3.

GIX outperformed all RE methods/models in Precision, Recall, and F-score for all four datasets: BioInfer, HPRD50, IEPA, and LLL. GIX produced a significant improvement of 12% in Precision on HPRD50 compared to the previous best model. BioBERT's improved performance in biological RE compared to traditional models like CNN and LSTM can be attributed to its pre-training on biological text, capturing contextual word representations, and transfer learning capabilities. The combination of BERN2's ability for normalization of named entities, along with the proposed anonymization of entities reduces sentence complexity without altering the lexical structure, and thus contributes to enhancing the model's accuracy of prediction. The robustness of the superior performance of GIX was further confirmed by its consistent performance across all four datasets.

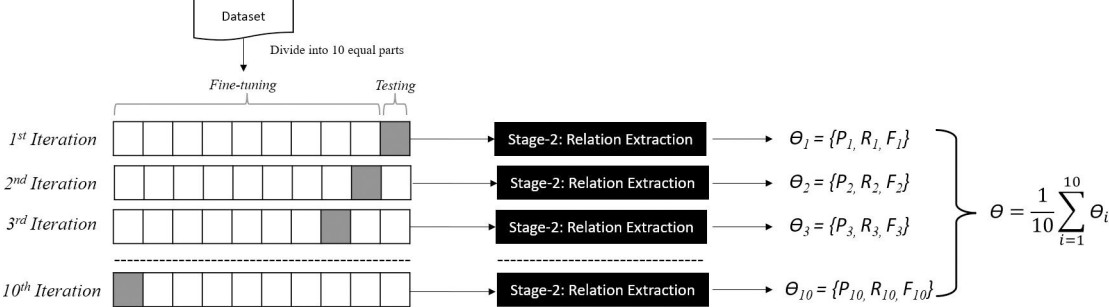

**Fig 7. The 10-fold cross-validation process for evaluating the relation extraction capability (Stage-2) of the GIX.** Here, each iteration consists of equally divided 10 folds of the dataset, where 9 folds (white blocks) are used for fine-tuning, and the 10th fold (grey block) is used for testing. The overall performance (Θ) is obtained by averaging the performance of each iteration (Θ$i$). The notations $P$, $R$ and $F$ represent Precision, Recall, and F-score.

**Table 3. Ten-fold cross-validation results (%) P: Precision; R: Recall; F: F-score.**

| Methods | BioInfer | | | HPRD50 | | | IEPA | | | LLL | | |
|---|---|---|---|---|---|---|---|---|---|---|---|---|
| | P | R | F | P | R | F | P | R | F | P | R | F |
| DNN [57] | 53.9 | 72.9 | 61.6 | 58.7 | 92.4 | 71.3 | 71.8 | 79.4 | 74.2 | 76.0 | 91.0 | 81.4 |
| Bi-LSTM [50] | 87.0 | 87.4 | 87.2 | - | - | - | - | - | - | - | - | - |
| RNN + CNN [58] | 56.7 | 67.3 | 61.3 | 69.6 | 82.7 | 75.1 | 64.3 | 65.8 | 63.4 | 72.5 | 87.2 | 76.5 |
| MCCNN [51] | 81.3 | 78.1 | 79.6 | - | - | - | - | - | - | - | - | - |
| GK [52] | 56.7 | 67.2 | 61.3 | 69.6 | 82.7 | 75.1 | 64.3 | 65.8 | 63.4 | 72.5 | 87.2 | 76.5 |
| CK [59] | 65.7 | 71.1 | 68.1 | 67.5 | 78.6 | 71.7 | 68.5 | 76.1 | 70.9 | 77.6 | 86 | 80.1 |
| NHGK [53] | 59.3 | 68.1 | 63.4 | 72.4 | 79.8 | 75.3 | 67.8 | 85.3 | 74.6 | 86.2 | 92.1 | 89.1 |
| EDG [54] | 57.6 | 59.9 | 58.7 | 69.9 | 76.2 | 72.9 | 76.7 | 83.3 | 79.9 | 92.1 | 78.2 | 84.6 |
| PIPE [42] | 68.6 | 70.3 | 69.4 | 62.5 | 83.3 | 71.4 | 63.8 | 81.2 | 71.5 | 73.2 | 89.6 | 80.6 |
| WWSK [55] | 61.8 | 54.2 | 57.6 | 66.7 | 69.2 | 67.8 | 73.7 | 71.8 | 72.9 | 76.9 | 91.2 | 82.4 |
| iLSTM+tAttn [41] | 88.9 | 89.3 | 89.1 | 78.6 | 78.7 | 78.5 | 81.7 | 82.3 | 81.3 | 84.8 | 84.3 | 84.2 |
| RCNN [56] | 87.4 | 86.5 | 86.9 | 74.9 | 82.8 | 77.7 | 71.6 | 80.6 | 75.5 | 80.5 | 87.2 | 83.2 |
| GIX RE | **91.1** | **92.9** | **92.0** | **91.5** | **93.3** | **92.2** | **89.4** | **89.5** | **88.9** | **93.9** | **92.4** | **93.9** |

## Exp2- comparison of GIX with manual curation of a benchmark dataset

The objective of this experiment was to evaluate the performance of GIX in capturing genetic relations against a manually curated benchmark dataset. We experimented using the known target network information for the LLL dataset. The other three datasets, IEPA, HPRD50 and BioInfer, contain generic interactions involving different organisms, including humans and model organisms, while the relations present in the fourth dataset, LLL, are confined to a single bacterial species, *Bacillus subtilis*. In GIX, fine-tuning is required for both the Sentence Eliminator 1 (Stage-1) and Relation Classification (Stage-2), as they use a variant of the BioBERT model. As given in Table 4, the fine-tuning was performed using sentences from the HPRD50, BioInfer, and IEPA datasets, while the testing was done using the independent LLL dataset. The LLL dataset contains sentences containing genetic interactions of type action, regulation, binding, and promotion of cell transcription activity in *B. subtilis*. The keywords used to extract information from PubMed about transcription in *B. subtilis* from the abstracts of published literature were "*Bacillus subtilis gene expression regulation transcriptional*". The maximum number of retrieved articles was set to 1000, so that only highly relevant papers were extracted.

The search for transcriptional regulations in *Bacillus subtilis* returned 371 abstracts containing 2,865 sentences. The Sentence Eliminator-1 and Sentence Eliminator-2 rejected 1,184 and 692 sentences, respectively, leaving 989 sentences for RE. The process extracted 1,120 relations from these sentences. Through the refinement step in the GIX post-processing stage, the extracted relations were further processed and condensed into 706 interactions (shown in Fig 8).

**Table 4. Dataset used for fine-tuning of BioBERT models for RE classification and sentence elimination 1.**

| Dataset | Experiments | | Relation extraction classification | |
|---|---|---|---|---|
| | Experiment with benchmark dataset (Exp2): LLL | Experiment real-world database (Exp3): RegulonDB | Positive | Negative |
| LLL | | x | 164 | 166 |
| IEPA | x | x | 335 | 482 |
| HPRD50 | x | x | 163 | 270 |
| BioInfer | x | x | 2000 | 2500 |

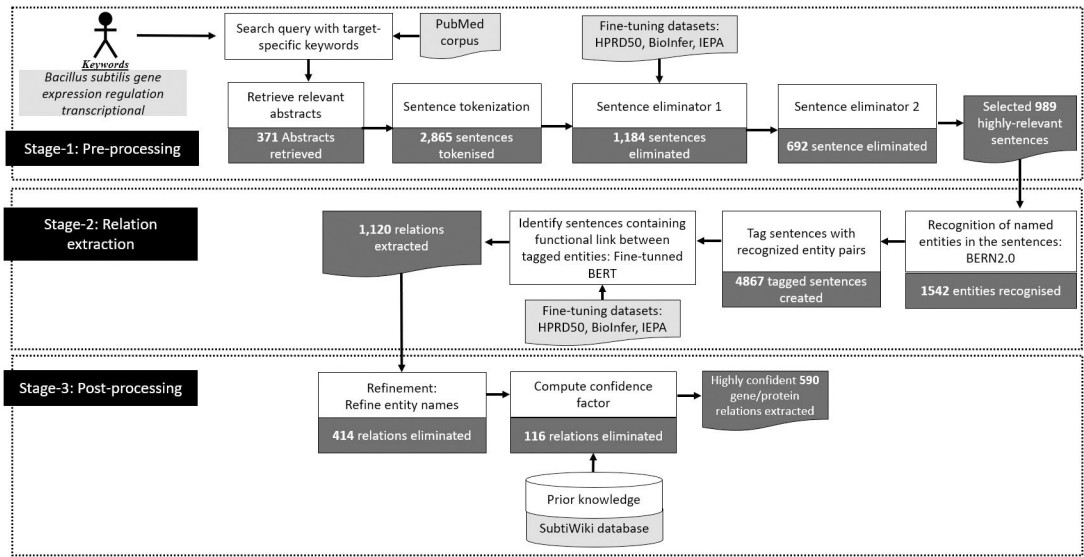

**Fig 8. GIX's extraction process for *Bacillus subtilis* relations illustrating outputs at each step (dark grey blocks) and inputs such as datasets, databases, or target-specific keywords (light grey blocks).**

To calculate $CF_{e_a e_t}$ using Eq 1, prior information about known regulators and regulations was obtained from Subtiwiki [60], a comprehensive online resource and database dedicated to the bacterium *Bacillus subtilis*. As depicted in Fig 9, we observed that the $CF_{e_a e_t}$ of the majority of relations when not considering prior knowledge, lay between 0.9 and 1. To achieve a balanced impact of prior knowledge and literature-based extraction on the overall CF, we set $K$ to

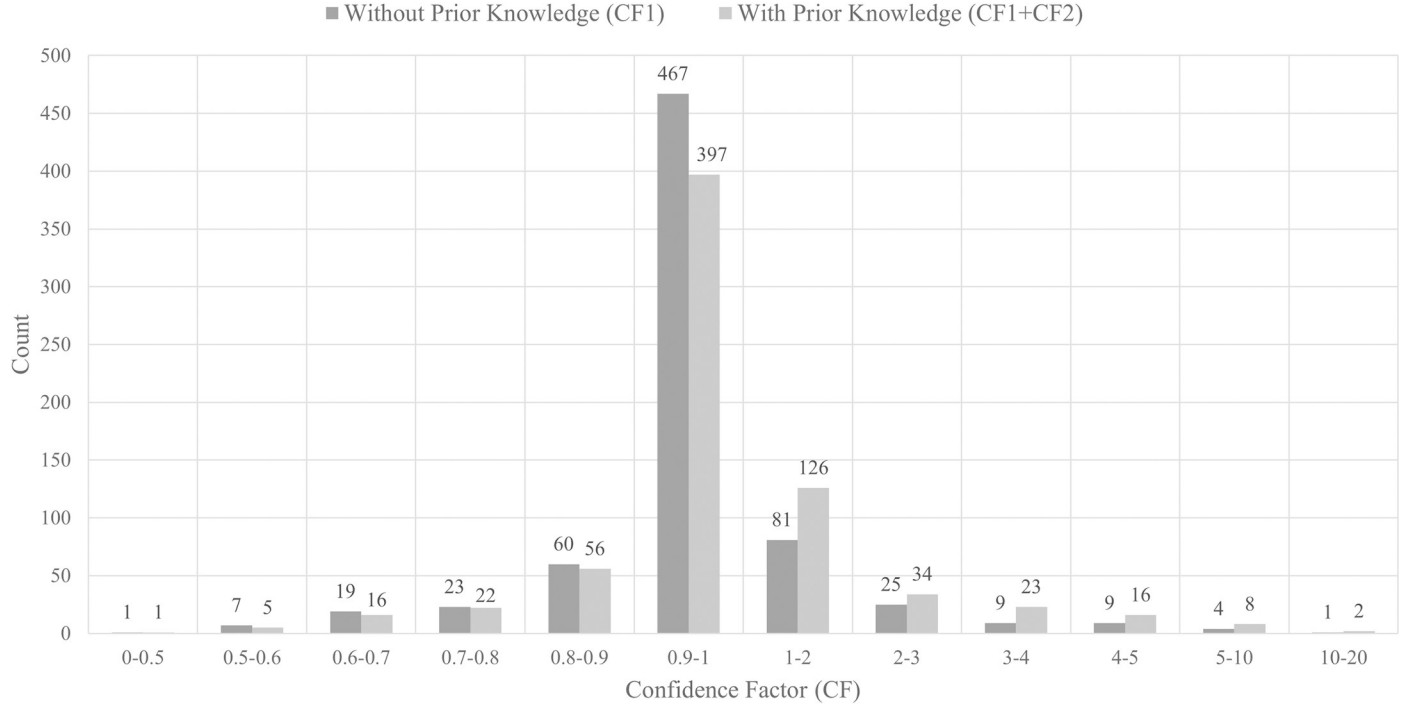

**Fig 9. Confidence factor ($CF_{e_a e_t}$) of the extracted relations for *Bacillus subtilis* distribution of $CF_{e_a e_t}$ with and without prior knowledge.**

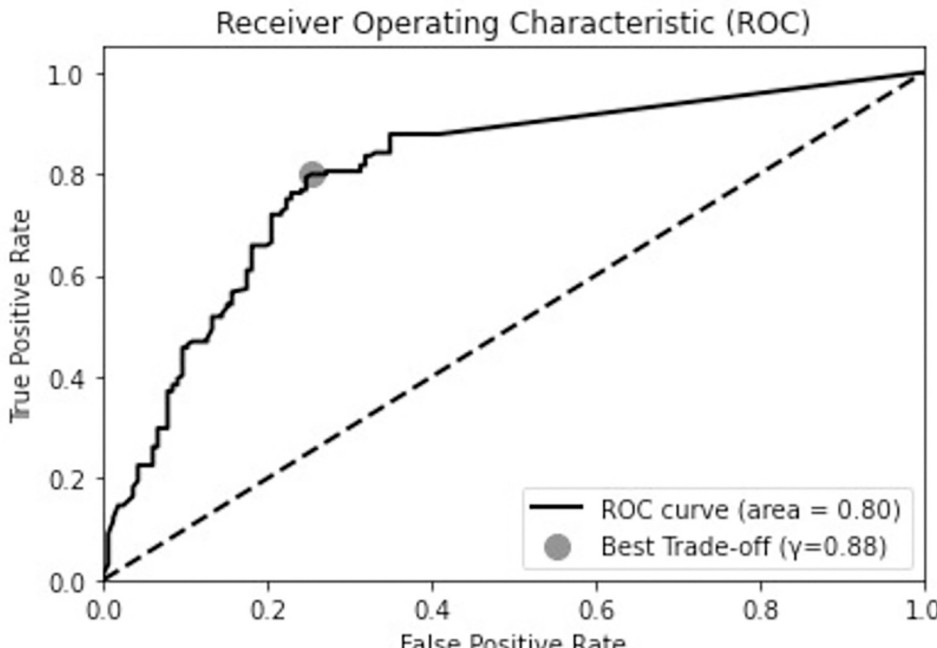

**Fig 10. ROC curve analysis used to determine the optimum threshold (γ) using relations extracted using GIX from 77 sentences in the LLL dataset.**

1. The impact of the $CF_{e_a e_t}$ is evident in Fig 9, which shows an increase in the number of relations within higher $CF_{e_a e_t}$ intervals with the activation of the prior knowledge component. After incorporating the prior knowledge component, the threshold value, γ, for $CF_{e_a e_t}$ was set at 0.88. This setting was experimentally determined after performing ROC curve analysis to identify the best trade-off between true positive rate and false positive rate (shown in Fig 10). With γ = 0.88, 590 interactions were extracted.

Table 5 displays the results of two experiments labeled as Exp2-(i) GIX-RE (only relation extraction model) and Exp2-(ii) GIX (Full Framework). In experiment Exp2-(i), only the relation extraction model of GIX, corresponding to Stage-2 of the framework, is utilised. The datasets used for fine-tuning of the relation classification model for experiment Exp2-(i) include HPRD50, IEPA, and BioInfer while LLL dataset is used for testing. In experiment Exp2-(ii), the full GIX framework is under investigation. Here, the fine-tuning datasets consist of HPRD50, IEPA, and BioInfer. While the testing dataset comprises of sentences extracted from PubMed related to *Bacillus Subtilis* regulatory interactions. These two experiments demonstrate that GIX, using just a few target-related keywords instead of the manually refined and formatted sentences was able to maintain a similar level of accuracy of extracted interactions without losing any of the sentences. The GIX framework is therefore robust, because, despite the elimination of a large number (2,109) of sentences during pre-processing and the loss of

**Table 5. Performance comparison of relation extraction from LLL sentences (i) using the GIX RE model and LLL sentences from the dataset, and (ii) using the GIX Framework and target-related keywords.**

| Method | Precision | Recall | F-score | Number of instances confirming true interactions found |
|---|---|---|---|---|
| GIX RE (LLL– 330 sentences) | 86.30 | 79.35 | 81.29 | 130 |
| GIX (Full Framework) | 86.30 | 79.35 | 81.29 | **255** |

almost half of the interactions due to post-processing refinement, there was no loss of information when using GIX. Furthermore, 130 out of 164 true interactions were confirmed 255 times by GIX by extracting the same interactions from multiple sentences, demonstrating its ability to leverage several sources for identifying true regulations. As a result, we augmented the dataset with an additional 125 sentences with labelled entities containing LLL's true regulations. This experiment also demonstrated GIX's proficiency in efficiently enriching benchmark datasets with sentences containing newly introduced terminologies, biological processes, and relationships.

We observed a difference between model accuracies when fine-tuning and testing sentences belong to the same dataset (LLL) and when the model was fine-tuned on non-LLL data (BioInfer, HPRD50, and IEPA) and tested on LLL. The disparities in accuracies can be attributed to differences in the fine-tuning data. The use of non-related datasets in Exp2 is realistic because in the real-world, often, the data is not always truly aligned with the training dataset. In past, the attempts by researchers at generalisation have been less successful with their accuracies decreasing significantly. For instance, in ([41], the model that was trained using only BioInfer and tested on LLL exhibited a low accuracy of 33.50%. In contrast, our proposed model has a Precision of 86.30% which is a slight decrease from Exp1 but still higher than other models. Combination of diverse non-related datasets (BioInfer, IEPA and HPRD50) has led to better generalisation and hence improved performance.

## Exp3- comparison of GIX with manual curation of a real-world database

We used *E. coli* interactions available in the database RegulonDB to evaluate GIX's ability to automatically extract TF-gene, TF-transcriptional unit, TF-operon, and TF-TF regulations. The extracted relationships are validated using known relations from RegulonDB, ensuring accuracy, and the overall confidence factor is adjusted based on this ground truth for evaluating GIX-based relationships against manually curated ones by RegulonDB. The database maintains references to articles for each curated interaction. The corpus neither records the article segment (such as Abstract, Introduction, or Conclusions) nor the sentences used to report the interaction. To evaluate the performance of GIX, we compared the accuracy of extraction of regulations by GIX with the RegulonDB regulations that had been curated from abstracts. We identified 578 unique interactions in 554 associated papers that mention the entities (gene/protein name) in at least one sentence of the abstract. For sake of convenience, throughout the paper, these 578 interactions are referred as abstract-level relations. The datasets used for fine-tuning the Sentence Eliminator– 1 (Stage-1) and the Relation Classification (Stage-2) were the four available benchmark datasets, BioInfer, HPRD50, IEPA and LLL. The testing dataset was formulated using sentences not found in the fine-tuning data sets. These sentences were extracted by GIX through keyword-based extraction from PubMed. The sentences comprising the testing dataset, obtained from published literature, are related to *E. coli*, while the benchmark datasets used for fine-tuning represent different domains. For example, the IEPA dataset is focused on biochemical relations and the LLL is dedicated to *Bacillus subtilis*. Thus, there is no overlap between the content of these four datasets and the *E. coli*-related sentences used for testing. The keywords used to extract the abstracts of published literature describing *E. coli* transcription from PubMed were "*E coli Escherichia coli gene expression regulation transcriptional*". To ensure the extraction of only highly relevant papers, the maximum number of retrieved articles was set to 1000. As in the previous experiment, we determined the threshold value $\gamma$ for CF, which was set at 1.5. The inputs and outputs of each process in GIX for Exp3 are depicted in Fig 11.

As depicted in Fig 12, in the absence of prior knowledge, the $CF_1$ associated with the majority of relations is distributed within the ranges [0, 2] and [4,10]. As per Eq 2, $CF_2$ will vary

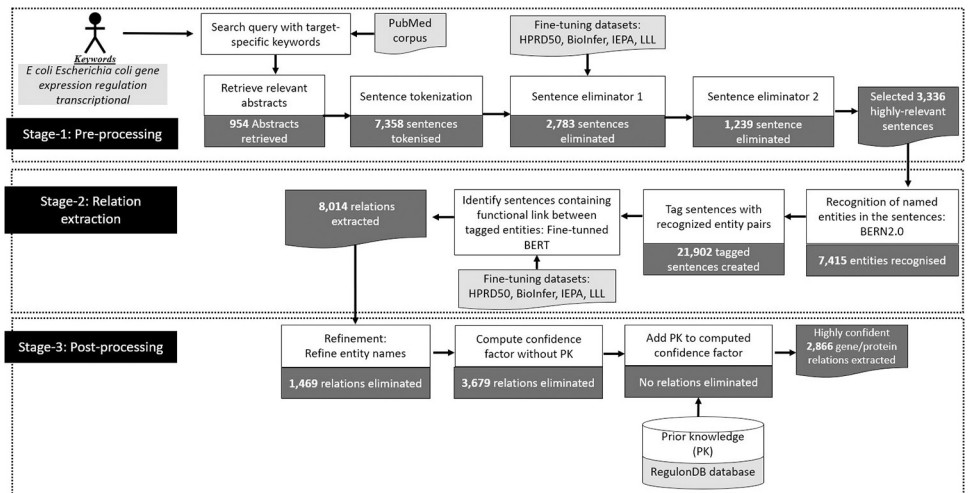

**Fig 11. GIX's extraction process for Escherichia coli relations illustrating the outputs at each step (dark grey blocks) and inputs such as datasets, databases, or target-specific keywords (light grey blocks).**

between 0 and 3. To balance the influence of prior knowledge and extraction based on existing literature on the cumulative CF, we set the parameter $K$ to 2, after averaging the higher values within the ranges of $CF_1$ and subsequently dividing by the maximum value of $CF_2$. To compute the $CF_{e_a e_t}$ value using Eq 1 and validate the extracted relations, we used the regulatory interactions TF-TF, TF-operon, TF-gene, TF-TU, and the regulators annotated in RegulonDB as prior knowledge.

We retrieved 954 abstracts containing 7,358 sentences. During pre-processing, 4,022 sentences were eliminated, leaving 3,336 sentences The RE stage extracted 8,014 positive entity pairs from within these 3,336 sentences. After post-processing refinement, we extracted 2,866 interactions. Upon evaluating the extracted relations against the 578 abstract-level interactions, 456 interactions were accurately identified. An additional 622 GIX extracted regulations were confirmed by relationships from RegulonDB that have not been annotated from abstracts of

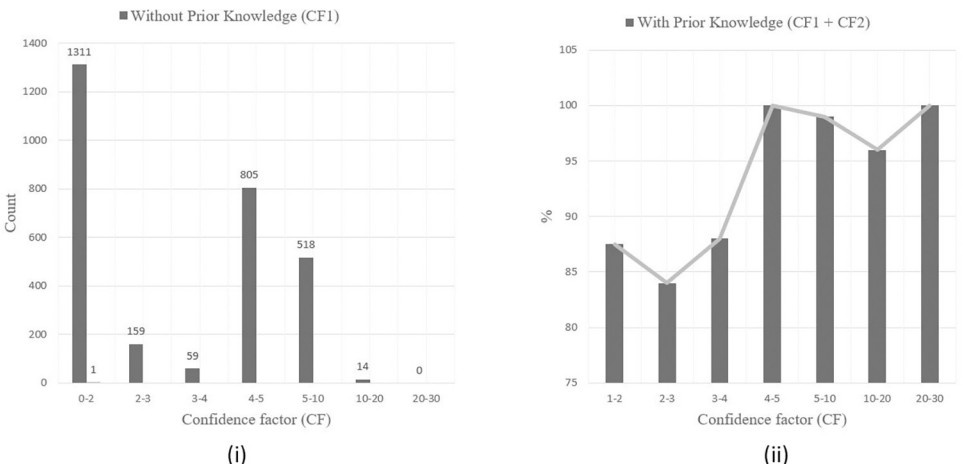

**Fig 12.** (i) Confidence factor ($CF_{e_a e_t}$) pertaining to the extracted relations for the *E. coli* distribution $CF_{e_a e_t}$ of without prior knowledge (PR) (ii) the percentage of relations extracted whose confidence factor $CF_{e_a e_t}$ was influenced by prior knowledge.

referenced literature. Of the regulations extracted by GIX, 92% were influenced by prior knowledge in their computed CF. Fig 12 shows that a significant number of relations with high confidence values are either known relations or have a known controlling entity. The interaction yielding the highest confidence value, of 24.65, was *CytR→CRP*. The high confidence value of the extracted relations provides a strong certainty in the accuracy and reliability of the extracted information.

## Application of GIX

Transcriptional regulation relations automatically extracted using GIX are important for understanding biological processes, particularly in the area of gene regulatory network inference. In this section, we demonstrate the way in which the output generated by GIX can serve not only as a basis for the inference of Gene Regulatory Networks (GRNs) but can also incorporate. $CF_{e_a e_t}$ to provide valuable information about the reliability of each interaction.

For GRN inference, we chose the top 500 GIX-extracted transcriptional regulations of *E. coli* obtained in Experiment Exp3. The selection was based on the $CF_{e_a e_t}$ assigned to each relation. We used Cytoscape [61], an open-source software platform, to visualize the network (Fig 13). With the confidence factor of each relation used as its corresponding weight, an arc appears thicker for higher $CF_{e_a e_t}$. The node size corresponds directly to the in-degree of the node. The GRN can help understand the complex regulatory mechanisms governing gene expression in *E. coli*. These networks can also serve as prior knowledge for reconstructing GRNs using more advanced computational methodologies and address the excessive computational overhead.

Biological circuits offer valuable insights into molecular-level interactions, especially within GRNs. From the presented GRN network *for E. coli*, entities CRP, *fnr*, *CytR*, *fis*, *MarA*, *csgD* stand out as the key regulatory genes, regulating 77 genes among themselves. Identifying controller genes is crucial as they govern gene expression, influencing cell state and offer help in developing targeted treatments for genetic disorders like cancer. GRNs exhibit sparsity, evident in the presented network where 334 genes exhibit only 500 interactions. Further, it may also be noted that due to the high cost of wet lab experiments to determine interactions, exhaustive exploration of relationships among thousands of genes becomes impractical. While significant efforts have been made to develop advanced computational methods for inferring relationships using gene expression data, yet the noisy and scarce nature of the data poses

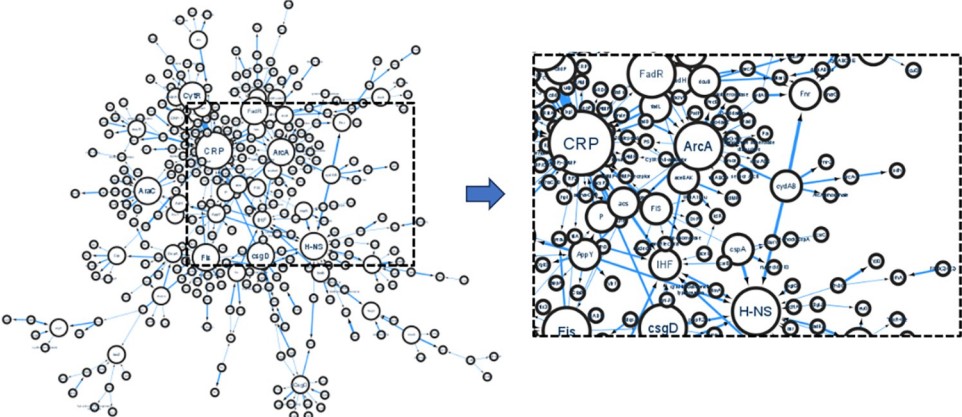

**Fig 13. Network diagram of 500 extracted *E. coli* gene/protein relations visualized using Cytoscape [61].**

several challenges in achieving improved accuracy and efficiency. Integrating relationships obtained via GIX as a-priori knowledge in the reconstruction process can significantly reduce convergence time, enhance overall accuracy, and facilitate the discovery of new relations. This iterative process, once refined, allows for cost-effective high-throughput experiments targeting specific entities, reducing the overall expense of uncovering crucial relationships.

## Conclusions

GIX (**G**ene **I**nteraction E**x**traction), our systematic and robust relation extraction framework, focuses on mining biological entity interactions from journal paper abstracts using domain-specific strategies and pre-trained attention-based models. The methodology underlying the GIX framework involves three stages: (i) pre-processing, (ii) relation extraction, and (iii) post-processing. The pre-processing stage uses a selection of keywords to obtain abstracts of highly relevant literature. Sentences that do not contain functional interactions and entity pairs are automatically excluded at this stage. In the relation extraction stage, the pre-trained large language models BERN2 and BioBERT are used for NER and RE. The associated entity-labelling schema reduces sentence complexity and improves model prediction accuracy. The post-processing stage refines the extracted relations by removing incorrectly recognised entities and assigns a novel confidence factor to quantify the correctness of an extracted relation. This confidence factor depends on both the information from multiple documents that corroborate a given regulatory relationships, and the pre-existing knowledge available from manually curated databases. The performance of GIX was validated using four benchmark datasets of gene/protein interactions. GIX's relation extraction ability surpassed the performance of previous state-of-the-art methods. GIX's performance against manually curated datasets and repositories was robust. We also observed the ability of GIX to augment existing datasets with new sentences from abstracts of published literature containing newly discovered terminologies and biological processes. The application of GIX to infer an *E. coli* gene regulatory network demonstrated its ability to work effectively with real world data.

Despite the rapid execution and high performance of pre-trained domain-specific large language models, the RE techniques described in the existing literature remain primarily confined to paper abstracts. The title, author list, affiliation, abstract, and keywords are easily available on Medline's PubMed repository. Designing a method to automatically extract text from the body of the paper may require source-specific code, authentication requirements and additional permissions to run data-scraping web services. Therefore, automatically extracting pieces of text from the body of the paper is relatively a more complex task than extracting text from abstracts. While the proposed method has focused on extracting relations using abstracts of publications, the approach is generic, and it can be easily extended for extracting relations using entire documents. For future work, we aim to extend GIX's applicability by seamlessly integrating extracted genetic relationships as prior knowledge for improved GRN reconstruction. Additionally, we will explore GIX's ability to identify multi-sentential relationships, providing a more comprehensive understanding of complex biological interactions.

## Author Contributions

**Conceptualization:** Jaskaran Kaur Gill, Madhu Chetty.

**Data curation:** Jaskaran Kaur Gill.

**Formal analysis:** Jaskaran Kaur Gill, Madhu Chetty.

**Investigation:** Jaskaran Kaur Gill.

**Methodology:** Jaskaran Kaur Gill, Madhu Chetty.

**Supervision:** Madhu Chetty, Suryani Lim, Jennifer Hallinan.

**Validation:** Jaskaran Kaur Gill.

**Visualization:** Jaskaran Kaur Gill.

**Writing – original draft:** Jaskaran Kaur Gill.

**Writing – review & editing:** Jaskaran Kaur Gill, Madhu Chetty, Suryani Lim, Jennifer Hallinan.

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
