## [Decision Letter · Decision Letter 0]

24 Nov 2023

PONE-D-23-31346Large language model-based framework for automated extraction of genetic interactions from unstructured dataPLOS ONE

Dear Dr. GILL,

Thank you for submitting your manuscript to PLOS ONE. After careful consideration, we feel that it has merit but does not fully meet PLOS ONE’s publication criteria as it currently stands. Therefore, we invite you to submit a revised version of the manuscript that addresses the points raised during the review process.

We look forward to receiving your revised manuscript.

Kind regards,

Michal Ptaszynski, PhD

Academic Editor

PLOS ONE

4. We are unable to open your Supporting Information file [S1 Supplementary data.zip ]. Please kindly revise as necessary and re-upload.

Reviewers' comments:

Reviewer's Responses to Questions

**Comments to the Author**

1. Is the manuscript technically sound, and do the data support the conclusions?

Reviewer #1: Partly

Reviewer #2: Partly

Reviewer #3: No

Reviewer #4: Yes

2. Has the statistical analysis been performed appropriately and rigorously? 

Reviewer #1: No

Reviewer #2: Yes

Reviewer #3: No

Reviewer #4: N/A

3. Have the authors made all data underlying the findings in their manuscript fully available?

Reviewer #1: Yes

Reviewer #2: No

Reviewer #3: No

Reviewer #4: Yes

4. Is the manuscript presented in an intelligible fashion and written in standard English?

Reviewer #1: Yes

Reviewer #2: Yes

Reviewer #3: No

Reviewer #4: No

5. Review Comments to the Author

Reviewer #1: This study presents a comprehensive framework for extracting gene regulatory information from biomedical literature using large language models. Their method consists of three main components: A Pre-processing unit that retrieves the relevant abstracts using keywords, a Relation Extraction unit that identifies the relations between entities, and a Post-Processing unit that ensures the overall quality of the extracted information through further clean-up. They show the effectiveness of their method using several popular datasets: Bioinfer, LLL, IEPA, and HPRD50. Their method noticeably outperforms all competing methods. This is a vital and interesting work. The manuscript is well-written, organized, and structured. However, this reviewer doubts the validity of the experimental setup and corresponding conclusions, as described below.

Major Comments:

1. It needs to be clarified if there is an overlap between the sentences in the training set and the articles/text used as evidence for gold-standard databases. If there is overlap, this creates bias due to information leakage, and a remedy needs to be applied.

2. Similarly, what about the overlap between sentences in/used in datasets such as BioInfer, etc, vs. RegulaonDB? If yes, this creates bias due to information leakage. Therefore, either a clarification is needed, or a remedy needs to be applied.

3. Additionally, the same datasets used for testing should not be used for finetuning LLMs as this introduces bias. A remedy must be applied to improve the reliability of the results.

4. It needs to be clarified how the hyperparameters, such as the number of epochs, were tuned. Was it 10-fold CV using only the train data? If not, there is bias. So, either a clarification is required, or a remedy is necessary to improve the reliability of the results. At the very least, 10-fold nested cross-validation must be used.

5. On a similar note, how were the rest of the decisions about the pipeline made? For example, finding optimal values for parameters like alpha and K. Was it only using training data? If not, there is bias. So, either a clarification is required, or a remedy is needed to improve the reliability of the results.

Minor Comments:

1. There are a few typos and grammatical errors. Please proofread or use an editing service like Grammarly.

2. The last sentence on page 6 requires a supporting citation.

3. In table 5, it is unclear whether the repeated performance for GIX-RE and GIX(full) is a typo or not.

4. Line 413 mentions "significantly". So, either provide p-values or avod using that term.

Reviewer #2: Dear Authors,

the manuscript definitely represents the interest for Plos One readership.

However, I have some comments and suggestions, which I ask you to address.

1. Please, specify more transparently what are the stages that are used for labelling sentences that do not include relations in the training set.

2. Please, discuss in the text, which parameters/models provide the highest accuracy of relations extractions in imbalanced data sets (scientific abstracts and publications are, by definition, imbalanced data sets).

3. How does the algorithm identify relations that are not provided in the sentences explicitly but can be supposed using the semantic analysis made by experts during manual annotation.

4. Please provide the code of the models in the Supplementary Materials.

5. I suggest that the authors discuss in more details the obtained results of gene regulatory network built using the developed approach from the biological point of view.

Reviewer #3: The manuscript presents the development of a tool for automatic extraction genetic entity pair interactions. The information is not well organized, each part is described in a very succinct way and there is the need to go forward and backwards to understand clearly the idea and purpose of each stage.

My main concern with this papers is related to originality. The paper presents some particular form of ensemble of several already developed tools, to produce another tool. The only aspects which has some originality is the fine tuning of BioBert, and this part, besides being barely explained, without details, seems to be wrong, as the same datasets used for test have been used in the fine tuning, thus providing biased results.

There is no source code or data made available, and due to this and the already mentioned lack of details regarding each part implementation, reproducing this work flow to use it or evaluate its characteristics would be almost impossible.

I think the work needs to be completely rewritten, starting from a clear description of the main contributions and why the authors consider it to be original,, and if the main objective is to produce a tool, give a clear and complete description of how to build and use the tool, preferably with available code to try it. Also there are many methodological issued in the experimental section that rise doubs in the validity of the results.

Detailed comments

- Sentence in lines 88-90 need to be revised. I think it should say “are due” instead of just due.

- Line 95. Says “BERT and ELMo, one of”, they are two, not one.

- Lines 116-117, it says “For the selection of abstracts, we use of task-specific…”, should be “… we make use of task-specific…”

- Line 215 on, section Preprocessing. It is not clear how the abstracts were pre-processed. Which search tool was used? How the keywords were used to filter the list of abstract candidates? Someone wanting to reproduce this work will need this information

- Line 227, Sentence Tokenization: It is not clear how Biobert and BERN2.0 are used to eliminate sentences that are not relevant. What do you input the models? Do you need to fine tune them also? How is this performed?

- Line 237, sentence eliminator 1: It is not clear how the fine tune is performed. And moreover, you are using the sentences in the 4 datasets you will use in the experiments to test the method, to fine-tune part of the method, in this way, the final test will not be independent of the training. Another concern, you need to explain better how the positive class for a sentence is decided. From your description, it seems that the relation must be encoded completely in one sentence, but sometimes an author would use several sentences to describe a positive interaction, how is this managed?

- Line 246, Sentence eliminator 2: Again, you discard a sentence if there are less than 2 gene/protein present in the sentence. But sometimes the relations are explained in several sentences, with one entity in each. And again, it is not explained at all how this was performed (and in this case it is not even mentioned which tool (perhaps BioBERT?) was used).

- Line 254, Relation Extraction: again, no details are given on how the fine-tuning of BERN2 and BioBert was done. Also, it is not clear to me why you need to replace the entity pair by generic labels. I imagine this can negatively affect the language model. The entity itself can carry context information useful for the language model. Imagine the word sequence “ The car is gray”, if I substitute car by a label $SUBSTANTIVE$ I am actually removing important information, as cars usually have a limited number of colors, and other substantives like “bears” have different sets of colors. So when you remove the real entity tag and replace it, you are removing contextual information useful for the language model. Thus, you need to clearly justify the need for this entity replacement.

- Line 320, equation (1): All variables should be defined immediately after the equation. It is difficult to understand several lines of discussion and explanation of why the equation will work without knowing what the variables are. For example, variable v is defined in line 327 (and is used ambiguously as in the equation it appears as v_s and in the explanation as just v)

- Line 344: it says that the value of alpha is set depending on hoy much emphasis is assigned to component P, but that is the role of constant K, which balances the important of the two terms, it is not clear why there is a need to set an extra parameter alpha that will be given later a value of 1 (without exploring its optimal value in any sense)

- Line 350: The same happens to constant K, which is later given a value of 1 without exploring optimal values.

- Line 356, it says “… are less likely to be true…” but this is the opposite as it is said before, having a higher CF gives more confidence in a relation, not lesser.

- Additional comment on CF. You should search for some ways to normalize this value, in other way there is no reference to which could be a good value of CF. It should be bounded between a minimum and a maximum known value. Otherwise you will find that for some particular use a CF=5 is indicative of a highly probable relation, and for another use the same value of CF=5 could indicate a low probability for a relation, which is very counterintuitive from the point of view of the user of your tool.

- Line 364. The numbering of experiments is quite confusing, as they are numbers 1.3, 1.4 and 1.5 , while they are not in section 1, and there is not 1.1 nor 1.2. You should label clearly your experiments, and then USE THOSE LABELS when you explain the experiment. As it can be seen in your manuscript, there is no reference to these labels later when the experiment are really explained.

- Line 383, section Selection of keywords: this procedure seems quite artisanal, needing manual intervention by the user. If you are going to use the most repeated keywords, you should automatically choose them, without having to curate them from a larger set.

- Line 398, Relation extraction capabilities (which seems to be experiment 1.3, but this is never mentioned): It is not clear how the validation experiments are performed. It says that the method was repeated 10 times using a different dataset for validation, but how exactly is this done? If there are 4 datasets and you leave one dataset out for validation, you can make only 4 repetitions. And also, it is not clear what is meant by training, what is trained and how? You should clearly explain how you run your experiments, for both, making it easy to reproduce your results, and checking if the experimental procedures was correct, or if you are using the same information for training and test. In the way it is written now, it makes impossible to reproduce the work, and rises concerns about the independence of the data used in training and test.

- Line 411, data on Table 3. It is not clear how the evaluation for alternative methods was obtained. Does the authors of the manuscript performed all test over the same datasets and in the same conditions for all the cases? Or they just take the value of some experiment in another work? If it is the second case the results are not directly comparable, as the experimental condition may have been different for each experiments, even using dependent data or overlapping data between train and test. This should be clarified very carefully and may invalidate the comparison.

- Another very important issue with this experiment is that Stage 1 of the proposed method includes the finetuning of BioBERT with the same 4 datasets used in this experiment to evaluate stage 2. In this way, even if the evaluation was done by leaving one dataset out (which was not clear, as already told two comments above), parts of the test set have been used for training (of stage 1), thus making invalid those results as the test dataset are no longer independent of the training set.

- Lines 442 to 445. The values for precision, recall and f-score in table 5 are exactly the same for the two reported methods. I am not sure if this is an error or effectively both alternatives produce the same values for those indexes. If this is the case, the text should discuss more clearly this issue and explain it.

- Line 445, figure 7: this figure shows the two terms used in calculation of CF. As already discussed, the values of CF should be normalized to a fixed range. And moreover, as can clearly be seen in figure 7, there is a very large difference in scale between both terms in equation 1, whit the second one clearly dominating by a large proportion over the first one. This makes the first term (which is the one were the proposed method contributes) completely useless, meaning that all the decisions will be directed by the previous knowledge, and in this way the full method is not taking any advantage from all the previously described work. What I mean is that if you completely remove the first term you can find relations and attribute them using the second, and the results will be no different of your proposal. In this way I think this experiment cannot demonstrate at all what it wanted to demonstrate.

- Lines 472 to 480. The discussion in this paragraph is hard to unde3rstand and should be rewritten. But I think I understand that it states that if you finetune properly the system without using any information from the test dataset (as I suggested in previous comments) then the performance of the system will be reduced. If this is what was meant, I must say this statement is obvious and is what I keep saying in my comments, that the validation was not done with an independent dataset, and thus is invalid. And if the performance is reduced, it means that the system cannot generalize, so it will be of no use to discover new relations, only to reinforce already known relations. I think this issue is key and should be clearly discussed and addressed, which was not properly done in this manuscript.

- Line 498. In the experiment reported in this area, the idea if I am not wrong is to use the proposed method to curate a database of interactions. What I find challenging and unexplained is why you use the same RegulonDB you are trying to curate as prior knowledge for the curation process. One would have expected some external knowledge as prior, so you can include external information that can incorporate new knowledge. This is like making yourself a test for your knowledge on a subject, and instead of asking someone else to correct your test to see if you answered it well, correcting it yourself, with the same knowledge you used to answer it. Key finding, you will get an A+ score, which does not mean you have acquired the knowledge.

For all those issued I consider this manuscript unsuitable for publication

Reviewer #4: To Authors:

General:

1.Overall text content should be revised, edited and shortened. Similarly the references which are relevant should be cited. Example: For such a very specific objective/aim of the paper, Introduction is too lengthy, more than 1400 words with 22 references.

2. Authors should keep in mind to revise to make each section concise and comprehensible.

Specific:

1.Introduction-to outline the Aims and Objectives of this paper for a reader.

2.Back ground literature should concentrate on current AI approaches which can facilitate data extraction beyond the Key words and subject headings(e.g. MeSH) for data mining.

3.Methods: should explicitly focus on pre-and post-processing, followed by the proposed confidence factor with examples from the published literature on the subject headings.

4.Results: the authors extend the argument with benchmark datasets and real-world database that GIX (Gene Interaction Extraction) should become the overarching approach. Is this the overall outcome derived?

5.Conclusion and Abstract should give homogenous statement (e.g. Sentence in Abstract, “With suitable experiments, we show GIX's capability to augment existing datasets with new sentences”).Thus GIX appears to be a tentative step.

6. PLOS authors have the option to publish the peer review history of their article (what does this mean?). If published, this will include your full peer review and any attached files.

Reviewer #1: No

Reviewer #2: **Yes: **Olga Tarasova

Reviewer #3: No

Reviewer #4: **Yes: **BIDHU K MOHANTI

---

## [Author Response · Author response to Decision Letter 0]

29 Jan 2024

We thank the editor and reviewers for their valuable insights and recommendations. All the suggested changes which were requested by the reviewers have been incorporated into the revised manuscript, thereby significantly improving its quality. 

In response to the four queries mentioned under “Journal requirements”, we need to inform that all these queries have been answered while addressing the reviewers. The supporting information file (S1 Supplementary Data) is now available through Github at https://github.com/JaskaranKaurGill1/Gene-Interaction-Extraction.git. Hence, the files should be accessible.

---

## [Decision Letter · Decision Letter 1]

26 Feb 2024

PONE-D-23-31346R1Large language model based framework for automated extraction of genetic interactions from unstructured dataPLOS ONE

Dear Dr. GILL,

Thank you for submitting your manuscript to PLOS ONE. After careful consideration, we feel that it has merit but does not fully meet PLOS ONE’s publication criteria as it currently stands. Therefore, we invite you to submit a revised version of the manuscript that addresses the points raised during the review process.

We look forward to receiving your revised manuscript.

Kind regards,

Michal Ptaszynski, PhD

Academic Editor

PLOS ONE

Reviewers' comments:

Reviewer's Responses to Questions

**Comments to the Author**

1. If the authors have adequately addressed your comments raised in a previous round of review and you feel that this manuscript is now acceptable for publication, you may indicate that here to bypass the “Comments to the Author” section, enter your conflict of interest statement in the “Confidential to Editor” section, and submit your "Accept" recommendation.

Reviewer #1: (No Response)

Reviewer #2: All comments have been addressed

2. Is the manuscript technically sound, and do the data support the conclusions?

Reviewer #1: (No Response)

Reviewer #2: Partly

3. Has the statistical analysis been performed appropriately and rigorously? 

Reviewer #1: (No Response)

Reviewer #2: Yes

4. Have the authors made all data underlying the findings in their manuscript fully available?

Reviewer #1: (No Response)

Reviewer #2: Yes

5. Is the manuscript presented in an intelligible fashion and written in standard English?

Reviewer #1: (No Response)

Reviewer #2: Yes

6. Review Comments to the Author

Reviewer #1: Authors have not submitted a detailed rebuttal letter and other required files. Therefore, this reviewer is unable to evaluate the revision.

Reviewer #2: Dear Authors,

Thank you for correcting the manuscript and addressing all my points in it.

I suggest that you include the main study design in the Methods or Results section in addition to Figure 1.

[Fig 1: Schematic of fully automated relation extraction using an LLM based GIX framework. The white

233 blocks signify the processes, and the grey blocks indicate the tools utilised within those processes.]

In particular, in addition to the schematic representation of relation extraction, it would be great to schematically represent all parts of the study design of all computational experiments used in the study, including "real-world applicability in inferring E. coli gene circuits".

Best Regards.

7. PLOS authors have the option to publish the peer review history of their article (what does this mean?). If published, this will include your full peer review and any attached files.

Reviewer #1: No

Reviewer #2: **Yes: **Olga Tarasova

---

## [Author Response · Author response to Decision Letter 1]

19 Mar 2024

We thank the editor and reviewers for their valuable insights and recommendations. All the suggested changes which were requested by the reviewers have been incorporated into the revised manuscript, thereby significantly improving its quality. We have addressed all reviewers' comments in detail in the attached 'Response to Reviewers.docx' document. 

It appears that there might be an issue with Reviewer 1 accessing the submitted rebuttal letter for Round 1. We want to emphasise that we have addressed all 30 queries raised by the four reviewers in detail (16 pages) with Reviewer 2 confirming satisfaction with our responses. However, for the convenience of Reviewer 1, we are resubmitting the round 1 rebuttal letter ‘Rebuttal Letter (Resubmit for Reviewer 1) - Previous Round.docx’ as Others. 

Response to Reviewer 2: Thank you for your constructive feedback and acknowledgment of the manuscript revisions. For demonstrating the real-world applicability of inferring E. coli gene circuits, we used Cytoscape (https://cytoscape.org/) to visualise the top 500 GIX-extracted transcriptional regulations of E. coli obtained from the experiment, Exp3. Thus, to illustrate the computational components of Relation Extraction experiments, we have incorporated schematic diagrams for the processes involved in experiments Exp1, Exp2, and Exp3 in Figure 7, 8 and 11 respectively. These diagrams are in addition to the schematic representation of GIX framework in Figure 1.

Refer to the attached 'Response to Reviewers.docx' document for further details.

---

## [Decision Letter · Decision Letter 2]

23 Apr 2024

Large language model based framework for automated extraction of genetic interactions from unstructured data

PONE-D-23-31346R2

Dear Dr. GILL,

We’re pleased to inform you that your manuscript has been judged scientifically suitable for publication and will be formally accepted for publication once it meets all outstanding technical requirements.

Kind regards,

Michal Ptaszynski, PhD

Academic Editor

PLOS ONE

Additional Editor Comments (optional):

Reviewers' comments:

Reviewer's Responses to Questions

**Comments to the Author**

1. If the authors have adequately addressed your comments raised in a previous round of review and you feel that this manuscript is now acceptable for publication, you may indicate that here to bypass the “Comments to the Author” section, enter your conflict of interest statement in the “Confidential to Editor” section, and submit your "Accept" recommendation.

Reviewer #1: All comments have been addressed

2. Is the manuscript technically sound, and do the data support the conclusions?

Reviewer #1: Yes

3. Has the statistical analysis been performed appropriately and rigorously? 

Reviewer #1: N/A

4. Have the authors made all data underlying the findings in their manuscript fully available?

Reviewer #1: Yes

5. Is the manuscript presented in an intelligible fashion and written in standard English?

Reviewer #1: Yes

6. Review Comments to the Author

Reviewer #1: The authors have done an excellent job at addressing all my comments and concerns. Thank you authors for your patience!

7. PLOS authors have the option to publish the peer review history of their article (what does this mean?). If published, this will include your full peer review and any attached files.

Reviewer #1: No

---

## [Editor Report · Acceptance letter]

30 Apr 2024

PONE-D-23-31346R2 

PLOS ONE

Dear Dr. Gill, 

I'm pleased to inform you that your manuscript has been deemed suitable for publication in PLOS ONE. Congratulations! Your manuscript is now being handed over to our production team.

Kind regards, 

on behalf of

Dr. Michal Ptaszynski 

Academic Editor

PLOS ONE